# Preoperative Multiparametric Quantitative Magnetic Resonance Imaging Correlates with Prognosis and Recurrence Patterns in Pancreatic Ductal Adenocarcinoma

**DOI:** 10.3390/cancers14174243

**Published:** 2022-08-31

**Authors:** Chao Qu, Piaoe Zeng, Hangyan Wang, Limei Guo, Lingfu Zhang, Chunhui Yuan, Huishu Yuan, Dianrong Xiu

**Affiliations:** 1Department of General Surgery, Peking University Third Hospital, Beijing 100191, China; 2Department of Radiology, Peking University Third Hospital, Beijing 100191, China; 3Department of Pathology, School of Basic Medical Sciences, Peking University Third Hospital, Peking University Health Science Center, Beijing 100191, China

**Keywords:** pancreatic ductal adenocarcinoma, multiparametric quantitative magnetic resonance imaging, histopathological techniques, prognosis, recurrence patterns

## Abstract

**Simple Summary:**

Magnetic resonance imaging (MRI) has been considered a noninvasive prognostic biomarker in some cancers; however, the correlation with pancreatic ductal adenocarcinoma (PDAC) remains inconclusive. The aim of our study was to identify quantitative MRI parameters associated with prognosis and recurrence patterns. In an analysis of data from the 136 patients ultimately included in this study, we found that the value of the pure diffusion coefficient D in intravoxel incoherent MRI is an independent risk factor for overall survival (OS) and recurrence-free survival (RFS), while a low value of D is significantly associated with a higher risk of local recurrence. All the patients have been operated on with histopathology for further evaluation. Based on the results of our research, we believe that it is possible in clinical practice to stratify patients based on quantitative MRI data in order to guide treatment strategies, reduce the risk of local tumor recurrence, and improve patients’ prognosis.

**Abstract:**

Magnetic resonance imaging (MRI) has been shown to be associated with prognosis in some tumors; however, the correlation in pancreatic ductal adenocarcinoma (PDAC) remains inconclusive. In this retrospective study, we ultimately included 136 patients and analyzed quantitative MRI parameters that are associated with prognosis and recurrence patterns in PDAC using survival analysis and competing risks models; all the patients have been operated on with histopathology and immunohistochemical staining for further evaluation. In intravoxel incoherent motion diffusion-weighted imaging (DWI), we found that pure-diffusion coefficient D value was an independent risk factor for overall survival (OS) (HR: 1.696, 95% CI: 1.003–2.869, *p* = 0.049) and recurrence-free survival (RFS) (HR: 2.066, 95% CI: 1.252–3.409, *p* = 0.005). A low D value (≤1.08 × 10^−3^ mm^2^/s) was significantly associated with a higher risk of local recurrence (SHR: 5.905, 95% CI: 2.107–16.458, *p* = 0.001). Subgroup analysis revealed that patients with high D and f values had significantly better outcomes with adjuvant chemotherapy. Distant recurrence patients in the high-D value group who received chemotherapy may significantly improve their OS and RFS. It was found that preoperative multiparametric quantitative MRI correlates with prognosis and recurrence patterns in PDAC. Diffusion coefficient D value can be used as a noninvasive biomarker for predicting prognosis and recurrence patterns in PDAC.

## 1. Introduction

Pancreatic ductal adenocarcinoma (PDAC) is an aggressive tumor, ranking fourth among cancer-related deaths in the world [1], and is predicted to become the second leading malignancy in cancer deaths in the United States by 2030 [2]. It has a poor prognosis, and the 5-year survival rate is of the lowest among all cancers [3]. At present, surgical resection is still the best treatment option for non-metastatic pancreatic cancer, but even if surgical resection is performed, recurrence after surgery is common, which adversely affects the long-term survival of patients. Therefore, it is necessary to carry out risk stratification and find a precise treatment for PDAC patients. Many studies investigating the risk factors for survival and recurrence in PDAC patients usually depend on postoperative pathological findings, such as tumor size, resection margin status, lymph node status, and differentiation grade [4,5]. At the same time, our previous study found that lymph node metastasis status was associated with the recurrence pattern of PADC [6]. If the risk factors for postoperative survival and the recurrence of PDAC can be determined before surgery in a non-invasive way, this will help to provide more precise and effective treatment strategies and earlier interventions (such as neoadjuvant therapy, extended surgical resection and postoperative adjuvant therapy, etc.) to reduce the risk of local tumor recurrence or distant metastasis and improve patient prognosis.

With the development of CT and MRI technology, the malignant degree of a tumor can be diagnosed and assessed preoperatively in a non-invasive way. MRI has advantages over CT in identifying tissue components, and functional magnetic resonance imaging (fMRI) can be quantitatively measured. Quantitative parameters such as the apparent diffusion coefficient (ADC) value measured by single-parameter functional diffusion-weighted magnetic resonance imaging (DWI) have been used in the diagnosis of a variety of tumors [7], monitoring tumor response to treatment [8,9,10], and predicting the degree of malignancy and prognosis [11]. With the continuous development of quantitative MRI techniques, multiparameter functional intravoxel incoherent motion (IVIM) DWI could provide quantitative parameters which could enable us to better understand the tumor microenvironment and the abundant extracellular matrix components of PDAC.

At present, there are few articles on the evaluation of survival and recurrence of PDAC using quantitative MRI, and as most of them are limited to the study of ADC value, no convincing conclusions can be drawn. Since our previous studies have shown that PDAC has different recurrence patterns, we intend to explore the correlation between preoperative multiparameter quantitative MRI and patient survival, prognosis or recurrence patterns, and further explore the correlation with the tumor microenvironment using histopathological methods.

## 2. Materials and Methods

### 2.1. Patients

In total, we collected 274 patients who received a radical resection for pancreatic tumors in the department of general surgery, Peking University Third Hospital, from June 2012 to December 2019. All the patients were pathologically diagnosed as PDAC postoperatively. The inclusion criteria included the following: (1) Complete data of enhanced CT and magnetic resonance examinations, including IVIM sequence, within 14 days before operation; (2) The solid part of the pancreatic mass was larger than 1.5 cm in diameter to ensure accurate MRI quantitative parameter measurements; (3) The MRIs were of good quality, without obvious artifacts and not seriously affected by other intra-abdominal organs; (4) Patients with no other severe heart, liver, lung, or kidney disease. The exclusion criteria included the following: (1) Neoadjuvant treatment before surgery; (2) Preoperative assessment of distant metastasis or inability to undergo radical surgical resection (R0/R1 resection); (3) History of other malignant tumors before surgery; (4) Death within 90 postoperative days or incomplete follow-up data. According to the inclusion and exclusion criteria, a total of 136 patients were included in this study (Figure 1). This retrospective study was approved by our institutional ethics review board.

### 2.2. Magnetic Resonance Imaging Techniques

MRI scans were obtained using the GE Discovery MR750 3.0T MRI scanner and 8-channel abdominal coil. The MRI protocol consisted of the axial T2-weighted image (repetition time/echo time (TR/TE), 12,000–15,000/81 msec; slice thickness, 6.0 mm; intersection gap, 1 mm; field of view, 32 × 32 cm; matrix, 320 × 320), and axial T1-weighted LAVA images (TR/TE 4/2.4 msec; slice thickness, 5.0 mm; field of view, 36 × 36 cm; matrix, 264 × 256). DWI was performed with an axial, free-breathing, single-shot echo-planar imaging with the following parameters: TR/TE, 15,000/70 msec; slice thickness, 6.0 mm; intersection gap, 1.0 mm; bandwidth, 250 kHz; field of view, 35 × 35 cm; matrix, 160 × 160; parallel imaging factor of 2; 8 b values (0, 25, 50, 100, 200, 500, 800, 1000 s/mm^2^).

### 2.3. DWI Analysis

The DWI data were postprocessed using built-in software (Functool MADC, GE) with the mono-exponential, bi-exponential, and stretched-exponential models, and ADC, D, D*, f, DDC, and α maps were calculated.

ADC was automatically calculated using the mono-exponential model with all b values:S(b) = S0 · exp (−b · ADC)(1)

The bi-exponential model—the mathematical relationship between b values and signal intensities—could be described using the following formula:S(b) = [(1 − f) · exp (−b · D) + f · exp (−b · D*)](2)
where D indicates unalloyed water molecular diffusion and f and D* are linked to the microcapillary perfusion effect, which may influence signal attenuation at low b values (b < 200 s/mm^2^). However, at high b values, the perfusion-related contribution to the signal attenuation becomes negligible; hence, true diffusion could be detected.

Finally, by fitting the stretched-exponential model, DDC and α were calculated as follows:S(b) = S0 · exp (−b · DDCα)(3)
where α varies from zero to one and characterizes the deviation of the signal decay from a mono-exponential decay. DDC indicates the composite ADC that is weighted by the volumes of water molecules with different diffusion coefficients.

All DWI-derived parameters were processed by two radiologists with 12 and 5 years of individual experience, who were blinded to the histopathological data. After independent image review, inter-observer agreement was evaluated and discordant results between the radiologists were adjusted by using consensus.

The region of interest (ROI) was manually drawn at the largest tumor cross section on the DWI (b = 1000) image, avoiding cystic degeneration, necrosis, and vessels. T2-weighted and contrast-enhanced images were used to guarantee the inclusion of the lesion. The ROI was automatically transferred to the ADC, D, f, D*, DDC, and α parametric maps. All DWI parameters in each ROI were calculated and the average over all 3 ROIs was used for analysis. The mean area of each ROI was taken to be 1 cm^2^ (range: 0.50–1.78 cm^2^).

### 2.4. Clinicopathological Findings

We obtained the basic clinical data, serological findings, and histopathological diagnosis of patients through our institutional database and electronic medical records. The clinicopathological parameters included in this study are as follows: age, gender, primary tumor location, tumor size, resectability, surgical procedure, preoperative carcinoembryonic antigen (CEA) and preoperative carbohydrate antigen 19-9 (CA19-9) value, CA19-9 value on the 7th day after surgery; histopathological grade, size, stage, presence of vascular invasion, perineural invasion, positive/negative resection margin, and lymph node involvement, etc. Patients were staged according to the 8th American Joint Committee on Cancer (AJCC) in the Cancer Staging Manual. The criterion for the surgical margin is that the tumor is more than 1 mm away from the margin under the microscope (R0), and the tumor cells are visible under the microscope within 1 mm from the margin (R1). The cut-off values for CEA (5 ng/mL) and CA19-9 (39 U/mL) are based on the standard levels used by our institution.

### 2.5. Follow Up

Patients have regular follow-up visits at our institution after discharge. CA19-9 and abdominal CT were examined every 3 months for the first 12 months and every 6 months thereafter. Other imaging studies, including MRI, bone scan, and positron emission tomography (PET-CT), were optionally performed when necessary. Overall survival (OS) was defined as the time from the date of diagnosis of pancreatic tumor by MRI examination until death or until the last follow up (31 December 2020). Recurrence-free survival (RFS) was defined as the interval between surgical resection and tumor recurrence. If no recurrence occurred, the RFS phase ended at death or last follow up. Imaging and serological examinations or surgical exploration were combined to determine the time between disease progression or disease recurrence during follow up. The recurrence patterns were defined as the location of the first recurrence, divided into two categories: local recurrence and distant metastasis. Local recurrence was defined as a tumor appearing in the retroperitoneal area, including in the operation bed, residual pancreatic tissue, or regional lymph nodes. Distant metastasis was defined as any recurrence in distant organs, including liver, lung, brain, and bone [12]. The chemotherapy regimens of patients after surgery were evaluated by professional doctors, and AG (Gemcitabine and Albumin Paclitaxel, 31 patients) or GS (Gemcitabine and S-1, 51 patients) regimens were used. The patients were followed up regularly.

### 2.6. Immunohistochemical Staining and Quantification

The mean time interval between MRI examination and surgery was 8 (range 1–14) days. All histopathological analyses were performed by two specialist pathologists with 15 and 5 years of individual experience, who were blinded to radiological and survival outcomes. Surgical specimens were serially sectioned for histology using 5 mm transverse sections oriented similarly to the axial MRI plane. MRI and histopathologic slides were reviewed independently, concordant location of the tumor was verified macroscopically and microscopically, and a representative block of tumor corresponding to the region analyzed by MRI was chosen for further histopathologic analysis [13]. Formalin-fixed and paraffin-embedded sections were cut to a thickness of 5 μm for hematoxylin-eosin and immunohistochemical staining. The degrees of cancer-associated fibroblasts (CAF) activation, angiogenesis, and tissue hypoxia in the tumor microenvironment (TME) were assessed through the α-SMA [14], VEGF [15], and HIF-1α anitbodies, respectively. Immunoreactivity was calculated by combining estimates of the percentage of immunoreactive cells (number fraction) and staining intensity (staining intensity fraction). For quantitative scores, 0%, 1% to 25%, 26% to 50%, and greater than 50% of positively stained cells were scored at 0, 1, 2, and 3, respectively. The staining intensity was rated on a scale of 0 to 3, where 0 is negativity, 1 is weak, 2 is medium, and 3 is strong positivity.

### 2.7. Statistical Analysis

The normality of MRI and histology data was tested using Kolmogorov–Smirnov (*p* > 0.05). Kaplan–Meier curves were calculated for OS and RFS for survival analysis, and differences were compared using the log rank test. The Cox proportional hazards regression model was used to analyze the independent risk factors associated with OS and RFS, and the clinicopathological data and quantitative magnetic resonance parameters that showed significant or possible significance with recurrence in the univariate analysis were included in the multivariate analysis, which was evaluated by a backward selection method to identify the independent risk factors associated with OS and RFS. The prognostic significance of covariates was expressed by hazard ratios (HR) with 95% confidence intervals (CI). Competing risks regression models were used to analyze recurrence outcomes. First recurrences in the form of local recurrence or distant metastasis were considered to be of competing risk based on their different therapeutic approaches and prognoses. Correlation studies between immunohistochemical (IHC) indices of histopathology and MRI multiparameters were performed using Pearson correlation coefficients. All reported *p* values were two-sided, and *p* < 0.05 was considered statistically significant. The optimal total point cut-off values of the quantitative magnetic resonance parameters in the survival analysis were determined by X-tile (version 3.6.1). Statistical analyses were performed using SPSS statistical software version 19.0 (winwrap basic) and Stata (version 14.0).

## 3. Results

### 3.1. Demographic and Clinicopathological Characteristics

As shown in Figure 1, a total of 276 patients in our center underwent pancreatectomy for pancreatic tumors and were pathologically diagnosed as having PDAC. A total of 140 patients were excluded from the cohort for the following reasons: 72 patients did not have complete enhanced CT and MRI data available for analysis within 14 days before surgery; 6 patients died within 90 days after surgery; 12 patients received neoadjuvant treatment before surgery; 4 patients underwent palliative surgery for locally advanced pancreatic cancer; 10 patients had a history of other malignancies within the last 5 years; another 36 patients (13.1%) were excluded because they were followed up for less than 12 months (no death or recurrence occurred during this period). The remaining 136 patients were included in the final analysis; their clinical–pathologic and radiologic characteristics are shown in Table 1. The average age of the patients was 65 ± 9.1 years. Among these patients, 62 were women and 74 were men. Surgical methods included 86 cases of pancreaticoduodenectomy (PD) (including 15 cases of vascular resection), 42 cases of distal pancreatectomy (DP) (including 2 cases of vascular resection), and 8 cases of total pancreatectomy (TP) (including 2 cases of vascular resection). A total of 82 patients received adjuvant chemotherapy after surgery.

### 3.2. Quantitative MRI Correlates with Survival Outcomes

The median follow-up period for this cohort was 21 (range 4–101) months. The median OS was 20 (95% CI:17.6–22.4) months, and the median RFS was 13 (95% CI:10.9–15.1) months. At the last follow up, 93 patients had died (68.4%) and 98 patients (72.1%) relapsed. Among them, the first recurrence manifested as distant metastasis in 59 patients (43.4%), whilst the manifestation of local recurrence occurred in 39 patients (28.7%). Among patients with distant metastases, liver metastases (n = 44, 32.4%) were the most common site.

The results of Kaplan–Meier curves survival analysis for the two groups are shown in Figure 2 and Figure 3. We found that a low preoperative ADC value (≤1.33 × 10^−3^ mm^2^/s), low D value (≤1.08 × 10^−3^ mm^2^/s), and high f value (>0.28) were all associated with lower RFS and OS rates (Figure 4 and Figure 5), with a median OS of 20 months for patients in the high-ADC value group compared to 13 months for patients in the low-ADC value group (Figure 2a), a median OS of 19 months for patients in the high-D value group compared to 25.5 months for patients in the low-D value group (Figure 2b), and a median OS of 23.5 months for patients in the low-f value group compared to 19.5 months for patients in the high-f value group (Figure 2c). In Cox multivariate regression analysis, low preoperative D value was an independent risk factor for the OS (HR: 1.696, 95% CI: 1.003–2.869, *p* = 0.049) and RFS (HR: 2.066, 95% CI: 1.252–3.409, *p* = 0.005) of patients treated with curative intent for PDAC (Table 2 and Appendix A). Other independent influencing factors associated with OS and RFS were tumor size and tumor margin status.

In addition, subgroup analysis of patients with different D and f values revealed that both OS and RFS were significantly higher in patients with high D and low f values (*p* = 0.001) (Figure 2d and Figure 3d). At the same time, in our analysis of the subgroup of patients treated with postoperative interventions, we found that patients in the high-D and high-f value group had significantly better outcomes with postoperative adjuvant chemotherapy (*p* = 0.034).

### 3.3. Quantitative MRI Correlates with Recurrence Pattern

In competing risks analysis, we found that patients with low preoperative D values had a higher risk of recurrence (SHR: 2.74, 95% CI: 1.801–4.179, *p* = 0.001); the median cumulative time to recurrence was 10 months in the low-D value group and 30 months in the high-D value group (Figure 6a). After controlling for other influencing factors, a low preoperative D value was significantly associated with a higher risk of local recurrence (SHR: 5.905, 95% CI: 2.107–16.458, *p* = 0.001; Figure 6b), and high preoperative D values showed a higher risk of distant recurrence (SHR: 1.70, 95% CI: 1.117–2.579, *p* = 0.013; Figure 6c).

In a subgroup analysis of patients with recurrence during follow up, we found that, among patients with distant recurrence, patients in the high-D value group who received postoperative chemotherapy showed significantly improved OS (*p* = 0.001, Figure 6d) and RFS (*p* = 0.003, Figure 6e). Statistical analysis was not performed in the local recurrence group due to the small number of patients with high D values.

### 3.4. Quantitative MRI Correlates with Histopathology

An immunohistochemical staining semiquantitative analysis of the tissue pathology specimens corresponding to the regions of interest on MRI from the 136 patients included in the study was performed to assess the correlation between the multiparametric quantitative MRI with neocollagenous fibers, angiogenesis, and tissue hypoxia in the tumor microenvironment of PDAC (Figure 4, Figure 5, Appendix A).

Through correlation analysis, we found that the α-SMA score was significantly negatively correlated with ADC value (*p* = 0.019) and D value (*p* = 0.000) and positively correlated with f (*p* = 0.001), the VEGF score was positively correlated with the f value (*p* = 0.020), and the HIF-1α score was negatively correlated with ADC value (*p* = 0.002) and D value (*p* = 0.000). At the same time, we found a strong positive correlation between α-SMA score and HIF-1α score (*p* = 0.018) and a strong positive correlation between D value and ADC value (*p* = 0.000). There is a strong negative correlation with the f value (*p* = 0.000) (Table 3 and Appendix A).

## 4. Discussion

In this study, we analyzed the correlation between preoperative quantitative multiparametric MRI and prognosis and pattern of recurrence in patients with PDAC treated with radical resection in a large sample size. The results of our study showed that a low preoperative ADC value, a low D value, and a high f value were all associated with lower RFS and OS rates. Patients with low preoperative D values had a higher risk of local recurrence and high preoperative D values had a higher risk of distant recurrence. Studies of correlation between multiparametric quantitative magnetic resonance imaging and patterns of recurrence after resection of pancreatic ductal adenocarcinoma are scarce and based on limited case series. Furthermore, through subgroup analysis, we also found a characteristic population that can significantly improve OS or RFS through postoperative adjuvant chemotherapy. At the same time, we found the degree of cancer-associated fibroblasts (CAF) activation, angiogenesis, and hypoxia of the tumor microenvironment can be reflected by multiparametric quantitative MRI.

The ADC value of single-exponential diffusion-weighted magnetic resonance imaging (DWI) models is a quantitative parameter used to assess water diffusion [16,17]. Several studies have shown that the ADC value in PDAC tumors is significantly lower than that in normal pancreas [18,19]; the main reason is that tumor-associated fibroblasts in the extracellular matrix of PDAC actively form dense collagen fibers, hindering the diffusion of water molecules [20]. Through immunohistochemical studies, we found a strong negative correlation between ADC value and α-SMA score, suggesting that ADC value may be affected by dense fibrosis, which is also consistent with the findings of Muraoka et al. [18], Ma et al. [21], and Wang et al. [22]. In addition, some studies have found that the ADC value can decrease with the increase in tumor cell density [23]. Several studies have previously confirmed the prognostic value of ADC values in many tumors [24,25,26], while some studies also reported the prognostic value of ADC values in PDAC [27,28,29,30]. Through correlation analysis, we found that the HIF-1α score was significantly negatively correlated with the ADC value (*p* = 0.002), tumor tissue hypoxia is an important feature of PDAC [31], tissue hypoxia is correlated with the degree of malignancy and invasiveness of PDAC [32]. Therefore, we believe that the reason for the low ADC value and poor prognosis of the patient is that the formation of dense fibrosis leads to significantly restricted molecular diffusion and severe tissue hypoxia. The findings of Mayer et al. [33] also confirmed our conjecture. However, because the ADC value is affected by many factors, such as the number of tumor cells, collagen fibers, and vessel density; it cannot specifically reflect the tumor microenvironment. We believe that the ADC value is an indicator of the degree of molecular diffusion in the overall responsive tumor tissue, and dense fibrosis may only be a very important factor affecting the ADC value. In the correlation analysis, we can also see that the correlation between ADC value and α-SMA score is lower than that between D value and α-SMA score. Similar results were similarly reflected in the multivariate regression analysis, which found a stronger correlation between ADC values and both OS and RFS in the univariate Cox regression analysis, but this correlation was manifested by the more significant D value in the multivariate Cox regression analysis.

Biexponential functional intravoxel incoherent motion (IVIM) diffusion-weighted magnetic resonance imaging (DWI)-based multi-b-value imaging method can provide multiparametric quantitative MRI, such as pure molecular diffusion coefficient (D), perfusion fraction (f), and perfusion-related diffusion coefficient (D*) for the specific analysis of molecular diffusion and microcirculation (perfusion) [34,35]. IVIM provided us with the possibility of further investigating the relationship between quantitative parameters of MRI and prognosis and the differences in the PDAC microenvironment. Our study found that the preoperative D value is very meaningful in predicting postoperative survival and recurrence, especially the recurrence pattern. In survival analysis, the low-preoperative-D value group showed significantly lower RFS and OS rates, which is consistent with the findings in a small sample study by Klaassen et al. [36], and we found that preoperative D value was an independent risk factor affecting OS and RFS.

Studies have shown that the D value reflects the diffusion related to the microstructure of tumor tissue, excluding the influence of factors such as vascular perfusion on the diffusion. Our study found a strong negative correlation between D value and α-SMA score (*p* = 0.000), which indicates that the degree of molecular diffusion is closely related to the degree of cancer-associated fibroblasts activation of the tumor microenvironment, implying that the denser neocollagenous fibers, the more restricted the diffusion of water molecules, leading to lower D values [13,37]. Interstitial fibrosis in PDAC is the predominant histopathological feature [18]; in the process of promoting the formation of fibrosis, cancer-associated fibroblasts of tumor stroma have the ability to promote local invasion [38] and increase cancer cell malignancy and therapeutic resistance [39,40]. Some studies have found that the local metastatic lymph nodes of PDAC usually have a higher degree of fibrosis [37]. Thus, we can argue that dense fibrosis increasing the cell density of tumor tissue resulted in limited molecular spread, decreased D value, and increased local invasive capacity of tumors. This hypothesis would explain why, in our experience, patients with a low D value showed worse OS and RFS and a high rate of local recurrence. Using preoperative multiparametric quantitative MRI, we could screen patients with low D values to target approaches throughout the course of treatment, such as adopting neoadjuvant treatment, and extending the extent of surgical resection or postoperative adjuvant chemoradiotherapy to reduce the risk of local recurrence of the tumor and improve patient outcomes. For patients with risk of distant recurrence in the high-D value group, postoperative adjuvant chemotherapy could be used to improve OS and RFS. The therapeutic strategy guided by the results of this study is expected to improve the overall prognosis of PDAC.

The perfusion fraction f value in IVIM sequences reflects the vascular distribution profile of PDAC and is considered useful for evaluating the blood perfusion levels of the tumor [41,42]. Our study found that patients with high preoperative f values showed poor OS and RFS, and that the f value had a positive correlation with VEGF scores (*p* = 0.020). Previous studies confirmed that angiogenesis in pancreatic ductal adenocarcinomas depends on the presence of angiogenic factors (such as vascular endothelial growth factor; VEGF) and is thought to be stimulated by hypoxia [15], which was consistent with our study. Through correlation analysis, we found that there was a significant negative correlation between D value and f value and HIF-1α score (*p* = 0.001). Therefore, we believe that the dense neocollagenous fibers or other components of the extracellular matrix surrounding cancer cells hinders the delivery of oxygen to cancer cells, which leads to tissue hypoxia and the formation of new capillaries. The results of the subgroup survival analysis showed that the patients in the high-D value and low-f value group had the best survival results, which further confirmed our findings. At the same time, in the subgroup analysis, we also found that, for the characteristic population with high D values and high f values, postoperative adjuvant chemotherapy can improve survival, which may be because chemotherapy can inhibit the formation of tumor angiogenesis, thereby improving prognosis. The mechanism remains to be further studied. In addition, although Klauß et al. found a good correlation between perfusion fraction f and microvessel density (MVD) in PDAC [43], and other studies have suggested that a high MVD is associated with a high risk of liver and lymph node metastasis, as well as a shorter survival time [44,45]. However, the correlation between VEGF expression and microvessel density (MVD) is still controversial, and this correlation may need to be confirmed by further research results.

For other quantitative MRI indexes that were measured, such as D*, DDC, and α values, this study failed to find a correlation with survival and recurrence in PDAC, and there is also no literature reporting the existence of such a correlation. However, some studies that did not find a correlation between f value and MVD [13] suggested that this might also be because the measurement of f values was far less stable than that of ADC values and D values, which might be related to the choice of b value in MRI, which led to conflicting results; the measurement of D* was the same for similar reasons [37,46].

This study has some limitations. First, since our study is a single-center retrospective study, there may be some biases (for example, lymph node positivity was not found to be an independent risk factor affecting prognostic survival, which might be due to selection bias or the sample size not being large enough). Multicenter, large-sample, randomized controlled studies are also needed to confirm our findings in the future. Second, to obtain a large sample size, our study had a large timespan, which may have led to some differences in the treatment strategy. Third, because IVIM images were acquired in a free breathing fashion, ADC and IVIM parameters may be affected by the variance caused by motion-induced dysregulation. However, because the pancreas is located in the retroperitoneum, the effects of respiratory motion may not be as great as they are in other abdominal organs [47]. Fourth, due to the complexity of the tumor microenvironment in PDAC, the results of this study (such as immunohistochemical indicators, etc.) cannot fully reflect the differences in the tumor microenvironment. Despite the limitations of this study, credible and meaningful research conclusions were obtained through research that was as scientifically reliable as possible.

## 5. Conclusions

In conclusion, preoperative multiparametric quantitative MRI correlates with prognosis and recurrence patterns in PDAC. A low preoperative D value was independently associated with lower RFS and OS and a higher postoperative local recurrence rate. The diffusion coefficient D value can be used as a noninvasive biomarker to predict prognosis and recurrence patterns and guide comprehensive PDAC treatment.

## Figures and Tables

**Figure 1 cancers-14-04243-f001:**
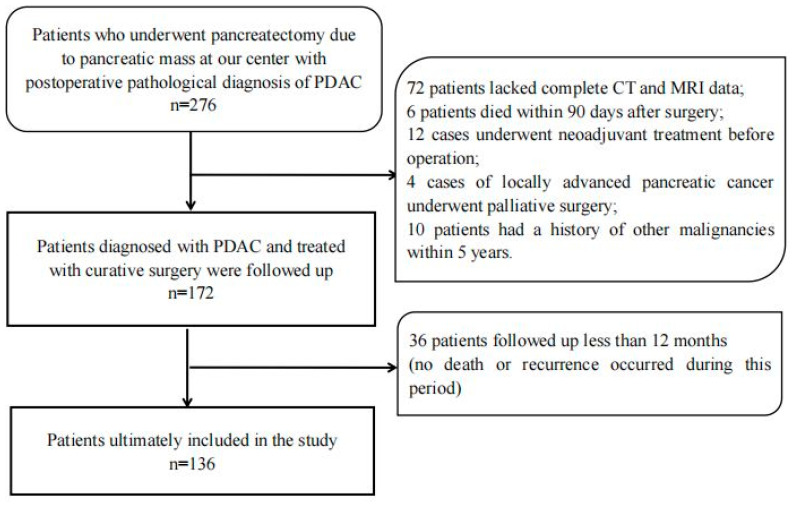
Description of study methods and demographic data.

**Figure 2 cancers-14-04243-f002:**
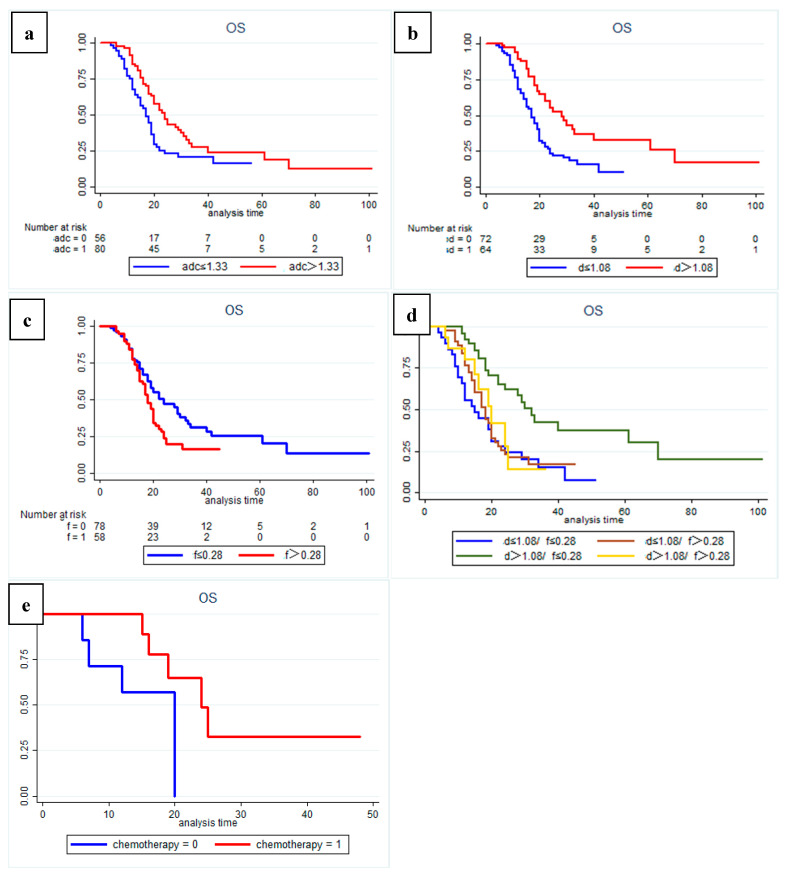
Kaplan–-Meier curves showing overall survival (OS) with clear differences between (**a**) high-ADC value group (sADC > 1.33 × 10^−3^ mm^2^/s) and low-ADC value group (sADC ≤ 1.33 × 10^−3^ mm^2^/s); (**b**) high-D value group (sD > 1.08 × 10^−3^ mm^2^/s) and low-D value group (sD ≤ 1.08 × 10^−3^ mm^2^/s); (**c**) low-f group (sf ≤ 0.28) and high-f group (sf > 0.28). (**d**) Subgroup analysis showed that patients in the high-D value combined with the low-f value group had a significantly better OS than patients in the other subgroups (*p* = 0.001). Additional subgroup analysis also identified (**e**) a significant positive effect of postoperative adjuvant chemotherapy on overall survival for patients in the high-D value combined high-f value group (*p* = 0.034).

**Figure 3 cancers-14-04243-f003:**
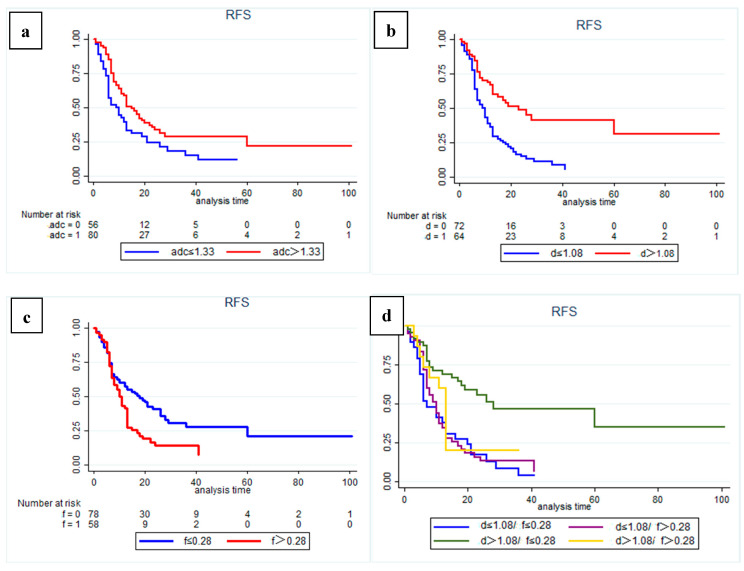
Kaplan–Meier curves showing recurrence-free survival (RFS) with clear differences in (**a**) high-ADC value group (sADC > 1.33 × 10^−3^ mm^2^/s) and low-ADC value group (sADC ≤ 1.33 × 10^−3^ mm^2^/s); (**b**) high-D value group (sD > 1.08 × 10^−3^ mm^2^/s) and low-D value group (sD ≤ 1.08 × 10^−3^ mm ^2^/s); (**c**) low-f group (sf ≤ 0.28) and high-f group (sf > 0.28). (**d**) Subgroup analysis showed that patients in the high-D value combined with low-f value group had a significantly better RFS than patients in the other subgroups (*p* = 0.001).

**Figure 4 cancers-14-04243-f004:**
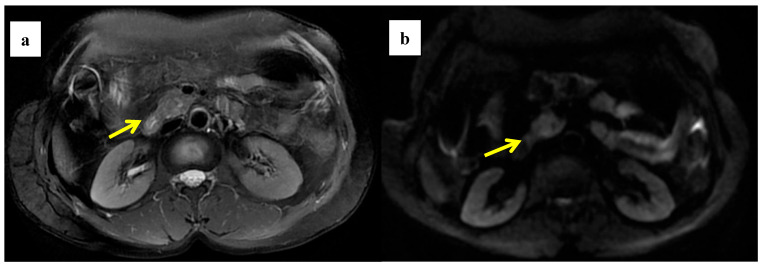
A 58-year-old woman was found to have a tumor with a head size of about 2.2 cm in the pancreas, which was pathologically confirmed to be pancreatic ductal adenocarcinoma after surgery; we found that the tumor was characterized by a high ADC value, high D value, and low f value after quantitative MRI measurement. She survived for 61 months without recurrence after surgery. MRI T2 (**a**); DWI (**b**); and ADC map (**c**) showed solid mass in the head of pancreas with measured ADC of 1.59 × 10^−3^ mm^2^/s, D value of 1.41 × 10^−3^ mm^2^/s, and f value of 0.18. Immunohistochemical staining showed that a few cancer-associated fibroblasts (CAFs) were present in the stroma ((**d**) α-SMA, ×10), carcinoma cells weakly expressed VEGF in the cytoplasm ((**e**) VEGF, ×10), and carcinoma cells showed weak immunoreaction of anti-HIF-1a in the nuclear((**f**) HIF-1α, ×20). These findings suggested that the tumor microenvironment was relatively quiet with a lower hypoxia level.

**Figure 5 cancers-14-04243-f005:**
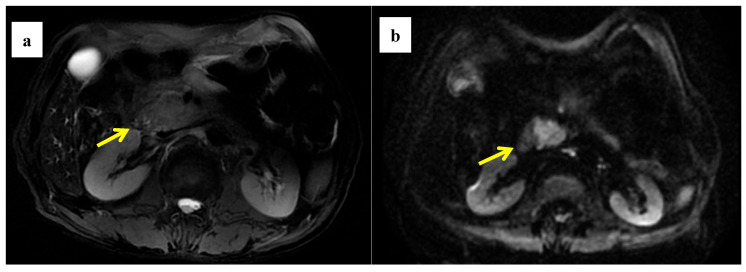
A 61-year-old man was found to have a tumor with a head size of about 2.1 cm in the pancreas, which was pathologically confirmed to be pancreatic ductal adenocarcinoma after surgery; we found that the tumor was characterized by a low ADC value, low D value, and high f value after quantitative MRI measurement. He was found to have a local recurrence at follow up 9 months after surgery and died 15 months later. MRI T2 (**a**), DWI (**b**), and ADC map (**c**) showed solid mass in the head of pancreas measured ADC measurements of 1.07 × 10^−3^ mm^2^/s, D value of 0.91 × 10^−3^ mm^2^/s, and f value of 0.47. Immunohistochemical staining showed an abundance of CAFs in the stroma ((**d**) α-SMA, ×10), carcinoma cells with higher expression of VEGF in the cytoplasm ((**e**) VEGF, ×10), and HIF-1a in the nuclear ((**f**) HIF-1α, ×20). These findings suggested that the tumor microenvironment was active with a higher hypoxia level.

**Figure 6 cancers-14-04243-f006:**
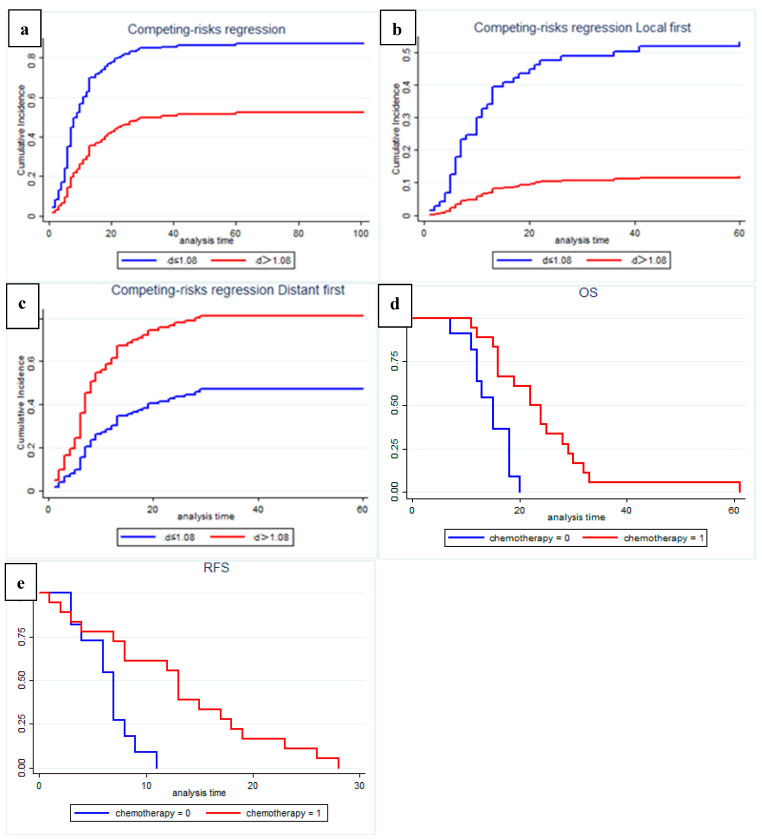
Recurrence risk and pattern analysis according to multiparametric quantitative MRI: (**a**) Fine and gray curve show risk of recurrence (*p* = 0.001); (**b**) Cause-specific analysis of competing risk of local recurrence pattern (*p* = 0.001); (**c**) Cause-specific analysis of competing risk of distant recurrence pattern (*p* = 0.001); (**d**,**e**) Among patients with distant recurrence, patients in the high-D value group who received postoperative chemotherapy may have significantly improved OS (*p* = 0.001) and RFS (*p* = 0.003).

**Table 1 cancers-14-04243-t001:** Clinical–Pathologic and Radiologic Characteristics of 136 Patients with PDAC.

Parameters	No. of Patients (%) (n = 136)
Age (years)	65 ± 9.1
Sex	
Male	74 (54.4%)
Female	62 (45.5%)
CEA (ng/mL)	
Normal (≤5)	97 (71.3%)
Elevated (>5)	39 (28.7%)
CA19-9 (U/mL)	
Normal (≤39)	22 (16.2%)
Elevated (>39)	114 (83.8%)
After surgery CA19-9 (U/mL)	
Normal (≤39)	69 (50.7%)
Elevated (>39)	67 (49.3%)
Type of surgery	
TP	8 (5.9%)
PD	86 (63.2%)
DP	42 (30.9%)
Tumor location	
Head	90 (66.2%)
Body	8 (5.9%)
Tail	38 (27.9%)
Tumor size (cm)	
≤2	22 (16.2%)
2–4	86 (63.2%)
>4	28 (20.6%)
N stage	
N0	76 (55.9%)
N1	48 (35.3%)
N2	12 (8.8%)
AJCC stage, 8th	
IA	15 (11.0%)
IB	50 (36.8%)
IIA	10 (7.4%)
IIB	48 (35.3%)
III	13 (9.6%)
Baseline resectability	
Resectable	91 (66.9%)
Borderline resectable	45 (33.1%)
Margin	
R0	94 (69.1%)
R1	42 (30.9%)
Nerve invasion	
(+)	17 (12.5%)
(−)	119 (87.5%)
Adjuvant Chemotherapy	
Yes	82 (60.3%)
No	54 (41.9%)
Recurrence	
Distant first	59 (43.4%)
Local first	39 (28.7%)
ADC (×10^−3^ mm^2^ /s)	
≤1.33	56 (41.2%)
>1.33	80 (58.8%)
D (×10^−3^ mm^2^ /s)	
≤1.08	72 (52.9%)
>1.08	64 (47.1%)
D* (×10^−3^ mm^2^ /s)	
≤7.94	40 (29.4%)
>7.94	96 (70.6%)
f	
≤0.28	78 (57.4%)
>0.28	58 (42.6%)
DDC	
≤1.5	90 (66.2%)
>1.5	46 (33.8%)
α	
≤0.86	105 (77.2%)
>0.86	31 (22.8%)

PD—pancreaticoduodenectomy; DP—distal pancreatectomy; TP—total pancreatectomy; CEA—carcinoembryonic antigen; CA19-9—preoperative carbohydrate antigen 19-9; ADC—apparent diffusion coefficient; D—pure diffusion coefficient; D*—perfusion-related diffusion coefficient; f—perfusion fraction; DDC—distributed diffusion coefficient; α—stretching coefficient.

**Table 2 cancers-14-04243-t002:** Cox Survival Analysis of Predictors of Overall Survival.

Variables	Category	Univariate Analysis		Multivariate Analysis	
		HR (95% CI)	*p* Value	HR (95% CI)	*p* Value
Age (years)	<65 vs. ≥65	0.777 (0.514, 1.176)	0.234		
Sex	Male vs. Female	1.167 (0.772, 1.765)	0.463		
CEA (ng/mL)	≤5 vs. >5	1.271 (0.715, 2.260)	0.415		
CA19-9 (U/mL)	≤39 vs. >39	1.068 (0.622, 1.835)	0.810		
After surgery CA19-9 (U/mL)	≤39 vs. >39	1.425 (0.941, 2.151)	0.094	-	
Type of surgery	PD	Ref			
	DP	1.197 (0.764, 1.875)	0.434		
	TP	1.060 (0.457, 2.460)	0.892		
Tumor location	Head	Ref			
	Body	0.637 (0.231, 1.758)	0.384		
	Tail	1.209 (0.767, 1.907)	0.414		
Tumor size (cm)	≤2	Ref		Ref	
	2–4	1.947 (0.994, 3.812)	0.052	1.845 (0.937, 3.630)	0.076
	>4	3.296 (1.551, 7.003)	**0.002**	2.559 (1.185, 5.524)	**0.017**
N stage	N0	Ref			
	N1	1.452 (0.945, 2.230)	0.088		
	N2	1.383 (0.674, 2.837)	0.377		
AJCC stage, 8th	IA	Ref			
	IB	1.986 (0.829, 4.761)	0.124		
	IIA	2.715 (0.824, 8.944)	0.101		
	IIB	2.625 (1.123, 6.263)	**0.026**		
	III	2.726 (0.989, 7.517)	0.053		
Baseline resectability	Resectable vs. Borderline	1.321 (0.869, 2.614)	0.231		
Margin	R0 vs. R1	1.594 (1.034, 2.457)	**0.035**	1.515 (0.970, 2.365)	0.068
Nerve invasion	(+) vs. (−)	1.057 (0.587, 1.905)	0.853		
Adjuvant Chemotherapy	Yes vs. No	0.686 (0.451, 1.043)	0.078		
ADC (×10^−3^ mm^2^/s)	>1.33 vs. ≤1.33	1.701 (1.123, 2.577)	**0.012**	1.336 (0.831, 2.149)	0.232
D (×10^−3^ mm^2^/s)	>1.08 vs. ≤1.08	2.189 (1.417, 3.381)	**0.001**	1.696 (1.003, 2.869)	**0.049**
D* (×10^−3^ mm^2^/s)	>7.94 vs. ≤7.94	1.070 (0.681, 1.682)	0.768		
f	>0.28 vs. ≤0.28	0.642 (0.424, 0.973)	**0.037**	0.859 (0.530, 1.393)	0.538
DDC	>1.5 vs. ≤1.5	1.299 (0.839, 2.010)	0.240		
α	>0.86 vs. ≤0.86	1.566 (0.922, 2.659)	0.097		

PD—pancreaticoduodenectomy; DP— distal pancreatectomy; TP—total pancreatectomy; CEA—carcinoembryonic antigen; CA19-9—preoperative carbohydrate antigen 19-9; ADC—apparent diffusion coefficient; D—pure diffusion coefficient; D*—perfusion-related diffusion coefficient; f—perfusion fraction; DDC—distributed diffusion coefficient; α—stretching coefficient.

**Table 3 cancers-14-04243-t003:** The correlation between quantitative MRI parameters and the histopathology of PDAC by Pearson correlation analysis.

	α-SMA		VEGF		HIF-1α		ADC		D	
	*r*	*p*	*r*	*p*	*r*	*p*	*r*	*p*	*r*	*p*
VEGF	0.017	0.849								
HIF-1α	0.207	**0.018**	−0.060	0.501						
ADC	−0.208	**0.019**	0.149	0.096	−0.320	**0.002**				
D	−0.475	**0.000**	−0.078	0.380	−0.304	**0.000**	0.512	**0.000**		
f	0.341	**0.001**	0.206	**0.020**	−0.096	0.282	0.129	0.147	−0.464	**0.000**

## Data Availability

The numerical datasets analyzed in the current study are available from the corresponding author on reasonable request. The Digital Imaging and Communications in Medicine (DICOM) files cannot be made freely available due to privacy restrictions.

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
