# Peer review of "Preoperative Multiparametric Quantitative Magnetic Resonance Imaging Correlates with Prognosis and Recurrence Patterns in Pancreatic Ductal Adenocarcinoma"

_cancers, 2022, doi:10.3390/cancers14174243_

Round 1
Reviewer 1 Report
Extensive language editing is required before final decision.
Reviewer 2 Report
Based on a cohort of 136 primary and borderline resectable, treatment-naïve ductal adenocarcinomas of the pancreas, the authors investigated whether multiparametric MRI correlates with survival (overall, recurrence-free) and pattern of recurrence (local, distant). Furthermore, MRI-based findings were correlated with the results of immunohistochemical investigation of the degree of fibrosis, vascularity and hypoxia in the tumour stroma. The results showed that the D value was an independent predictor of overall and recurrence-free survival as well as the pattern of disease recurrence.
The aim of this study is no doubt interesting and of high clinical relevance. However, I have concerns regarding the immunohistochemical investigations of tumour tissues:
- aSMA: is a marker of activated pancreatic stellate cells, which may be highly expressed in cell-rich collagen-poor stroma. Its expression level is often low in stroma with high collagen content. For the evaluation of fibrosis, the presence of collagen should be assessed, not of stromal cells.
- VEGF: the expression level of this endothelial growth factor does not necessarily reflect the degree of vascularisation. For the assessment of the latter, microvessel density should be measured based on immunohistochemical staining of an endothelial marker.
- HIF1a: the authors assessed HIF1a expression of the stroma. It is not clear (i) why the stroma was examined (and not the cancer cells), (ii) which cells in the stroma were assessed (activated stellate cells?) and (iii) how these stromal cells were identified and distinguished from other cells in the tumour stroma.
- Were the immunohistochemical stains done on a single or on multiple blocks? It is stated that the immunohistochemical assessment was spatially matched with the regions of interest on MRI. This is a difficult undertaking, but unfortunately, there is no information on how imaging and histopathology were aligned.
- Figures 4/d-f and 5/d-f: the images are of poor resolution, and the legends do not indicate which antibodies were used. Scale bars are missing. Some of the images show considerable background staining, for instance in Langerhans islets. While difficult to discern, immunostaining for HIF1a - a transcription factor - seems to be cytoplasmic rather than nuclear.
- The results of the semiquantitative assessment of the various immunohistochemical stainings are not provided.
Further comments:
- Table 1: the study series shows an unusually high N0-rate. Furthermore, N-stage was not found to be an independent prognostic factor, which stands in contrast with the fact that according to the literature, N-stage is a strong predictor of outcome. The authors may wish to comment on that.
Round 2
Reviewer 2 Report
The authors essentially explained why they choose the selected immunohistochemichal markers but did not make any of the suggested changes. Neither have they included any of the argumentation in support of the selection of their markers. It is not acceptable to use alpha-smooth muscle actin also as a marker for collagen deposition: this is clearly incorrect from a biological point of view. The same applies to using VEGF as a short-cut for microvessel density. The authors admit to the fact that the low lymph node metastasis rate is unusual, but - understandably - cannot change this. Part of the reasoning is entirely beside the point (eg. the surgeon staining the specimen surface is irrelevant to the fact that only one tissue block was investigated).
Round 3
Reviewer 2 Report
It is a fundamental principle of scientific research that if a particular feature is to be investigated, that very feature should be evaluated directly, or, should that be impossible, by a proxy that is directly correlated with the feature of interest.
Hence, investigation of the collagen content in cancer tissues requires the evaluation of collagen. While activated a-SMA-positive pancreatic stellate cells are the main source of collagen, (1) the former do not always produce collagen, (2) collagen production may be in highly varying quantities, and (3) stroma with a high collagen content may contain very few activated a-SMA-positive pancreatic stellate cells. Therefore, evaluation of a-SMA cannot be used as a substitute for the evaluation of collagen. The authors write that they have supplied their investigations with immunohistochemical detection of collagen 1. Unfortunately, the results are not included in the Results section of the manuscript, and neither is the Methods section updated (Which antibody? Dilution? Semiquantative scoring?). Instead, two images are shown as supplementary material. Unfortunately, in Fig. 2S, the area that in the H&E-stained section contains much fibrous stroma is not represented in the picture that shows the result of immunohistochemical staining, as there is a hole in the tissue exactly at that focus. The microphotographs showing immunohistochemical detection of a-SMA (Fig. 4) in the manuscript have remained unaltered. However, changes have been made to the legend of Figure 4, and these are incorrect as the photographic evidence provided in the figure does not support the statement “Immunohistochemical staining showed that the cancer-associated fibroblasts (CAFs) in the stroma and CAFs were the producing-cells of neocollagenous fibres (d, a-SMA, x10)”. Without staining for collagen, the production of collagen cannot be assessed – let alone the production of “neocollagen”. The addition to the legend stands in stark conflict with the above considerations. The authors claim to have “rewritten this section and related content in the discussion section”, but there are no track changes that indicate such a revision
Similarly, if perfusion in the tumour - as assessed on imaging - is to be correlated with perfusion in the cancer tissue, the microvessel density is to be investigated. While VEGF is indeed an angiogenic factor, its mere presence (detected immunohistochemically in this study) does not necessarily reflect the microvessel density, as there are a range of anti-angiogenic factors that may be secreted in the TME and may reduce the effect of VEGF (and are not investigated in this study). As the authors have not assessed microvessel density, the study does not provide evidence that can be correlated with tumour perfusion on imaging. The inclusion of two microphotographs showing CD31-immunostaining (Figs 2S & 3S) may do as an illustration, but does not qualify as results of a systematic investigation. The authors have added the following comment to the Discussion “But some studies yielded controversial data about the correlation of VEGF expression and microvessel density (MVD), so this correlation needs to be confirmed by further research results.”, but unfortunately, they have not included references for these studies to support their statement.
Taken together, the authors have not adequately addressed the serious concerns that were raised.
Round 4
Reviewer 2 Report
The authors have accepted the point that α-SMA is not a substitute for collagen deposition and that VEGF does not stand for microvessel density, and they have revised the text accordingly. There are, however, to statements in the Discussion in which conclusions are not supported by evidence:
- page 20 - “... we found a strong negative correlation between ADC value and α -SMA score, confirming that ADV value is affected by dense fibrosis”: here the correct wording would be “… suggesting that ADV value may be affected by dense fibrosis”. The authors have no evidence that tells which of the extracellular matrix compounds affect(s) the ADC value.
- page 20 – “… this indicates that the degree of molecular diffusion is closely related to the degree of cancer-associated fibroblasts activation of the tumor microenvironment, and the denser neocollagenous fibers are, the more restricted the diffusion of water molecules, leading to lower D values”: here the correct wording would be “… fibroblasts activation of the tumor microenvironment, implying that the denser neocollagenous fibers are, ….”.
- page 21 – “Therefore, we believe that the dense neocollagenous fibers surrounding cancer cells hinders the delivery of oxygen to cancer cells”: the authors present no data in support of this. Moreover, there are other components of the extracellular matrix, in particular hyaluronan, that are known to hinder diffusion of oxygen (and drugs) through the TME. A more correct wording would therefore be “… we believe that the dense neocollagenous fibers or other components of the extracellular matrix surrounding cancer cells ….”.
The fact that the use of α -SMA and VEGF does not allow assessment of collagen deposition and microvessel density and therewith more direct correlation with the quantitative MRI indexes should be added to the limitations of the study.
